# Genome-Wide Identification and Characterization of the *PP2C* Family from *Zea mays* and Its Role in Long-Distance Signaling

**DOI:** 10.3390/plants12173153

**Published:** 2023-09-01

**Authors:** Huan Wu, Ling Zhu, Guiping Cai, Chenxi Lv, Huan Yang, Xiaoli Ren, Bo Hu, Xuemei Zhou, Tingting Jiang, Yong Xiang, Rujun Wei, Lujiang Li, Hailan Liu, Imran Muhammad, Chao Xia, Hai Lan

**Affiliations:** 1Maize Research Institute, Sichuan Agricultural University, Chengdu 611130, China; 2019113006@stu.sicau.edu.cn (H.W.);; 2Department of Chemistry, Punjab College of Science, Faisalabad 54000, Pakistan; 3State Key Laboratory of Crop Gene Resource Exploration and Utilization in Southwest China, Sichuan Agricultural University, Chengdu 611130, China

**Keywords:** maize, PP2C, gene family, long-distance signaling, nitrogen

## Abstract

The protein phosphatase 2C (PP2C) constitutes a large gene family that plays crucial roles in regulating stress responses and plant development. A recent study has shown the involvement of an *AtPP2C* family member in long-distance nitrogen signaling in Arabidopsis. However, it remains unclear whether maize adopts a similar mechanism. In this study, we conducted a genome-wide survey and expression analysis of the PP2C family in maize. We identified 103 *ZmPP2C* genes distributed across 10 chromosomes, which were further classified into 11 subgroups based on an evolutionary tree. Notably, cis-acting element analysis revealed the presence of abundant hormone and stress-related, as well as nitrogen-related, cis-elements in the promoter regions of *ZmPP2Cs*. Expression analysis demonstrated the distinct expression patterns of nine genes under two nitrogen treatments. Notably, the expression of *ZmPP2C54* and *ZmPP2C85* in the roots was found to be regulated by long-distance signals from the shoots. These findings provide valuable insights into understanding the roles of *ZmPP2Cs* in long-distance nitrogen signaling in maize.

## 1. Introduction

Maize (*Zea mays* L.) holds significant global importance as a cereal crop, serving not only as a source of human food but also as a valuable resource for livestock feed and industrial raw materials [1]. Nitrogen (N) is a crucial macronutrient essential for plant growth and crop yield, with its availability and distribution within the soil exerting notable impacts. Additionally, nitrogen acts as a pivotal signaling molecule, influencing diverse biological processes in plants, such as lateral root growth and resistance to biotic and abiotic stresses [2,3,4,5]. Plant-accessible nitrogen predominantly exists in either inorganic or organic forms, with nitrate serving as the primary source under field conditions [6]. However, the uneven distribution of nitrogen in the soil due to rain erosion adversely affects plant growth and crop yield [7]. The extensive utilization of nitrogen fertilizers, aimed at increasing crop yield, has unfortunately resulted in environmental degradation [8]. Consequently, there is an urgent need to limit the application of chemical fertilizers while simultaneously enhancing overall nitrogen use efficiency without compromising crop production [9].

Due to the complexity of the soil environment, plants have evolved complex molecular mechanisms that enable them to regulate the efficiency of root nitrogen acquisition in response to fluctuations in the external nitrogen environment. This regulation ensures the fulfillment of aboveground nitrogen demands [10]. Two types of responses are involved: local nitrogen signaling and long-distance nitrogen signaling [11]. Specifically, when one side of the root system experiences local nitrogen deficiency, it triggers upregulation of nitrate uptake on the opposite side of the root system [12]. Recent studies have revealed that plants employ long-distance signals to maintain nitrogen homeostasis, especially when soil availability is low or nitrogen demand on the ground is high [13]. When plants are exposed to heterogeneous nitrogen conditions, the two parts of the roots communicate with each other via a root–shoot–root signaling pathway. This pathway involves *CEP-CEPR-CEPD1/CEPD2*, which facilitates plant nitrate absorption and utilization, thereby enhancing nitrogen use efficiency [14,15,16]. Moreover, when the external nitrate levels are sufficient or moderate, the plants induce the expression of *cepd-like2* (*CEPDL2*) to enhance the absorption of NO_3_^−^ via NRT2.1 [17]. NRT2.1, a key nitrate transporter, plays a central role in high-affinity nitrate uptake in roots and is activated at the post-translational level in response to nitrogen (N) starvation [18].

Post-translational modification of proteins through the action of kinases or phosphatases is an important mechanism that activates or inactivates signaling intermediates by adding or removing a phosphate group. These modifications are crucial in various physiological, developmental, and stress-tolerance mechanisms [19]. In Arabidopsis, a recent study discovered that a gene was upregulated under N starvation conditions [18]. This gene belongs to the *PP2C* (protein phosphatase 2C) gene family. *PP2C* has been widely implicated in controlling various stress responses and developmental processes in plants [20,21].

*PP2Cs* is one of the most important classes of phosphatases in plants, comprising 60–65% of all phosphorylases [22,23]. *PP2Cs* have diversified from prokaryotes to multicellular eukaryotes in terms of the number of genes and clades [24,25]. As a key phosphatase in plants, *PP2Cs* play important roles in regulating plant growth and biotic and abiotic stresses [21,26]. Numerous studies have shown that some *PP2C* genes are involved in the regulation of abscisic acid (ABA) signaling pathways in response to abiotic stress in different plant species, including *Zea mays*, *Arabidopsis thaliana*, and *Brachypodium* spp. [27,28,29,30]. These findings underscore the diverse functions of *PP2Cs* across different plant species within stress signaling pathways.

Early studies have shown that changes in gene activity in response to nitrate treatment require the activities of kinases and phosphatases [31]. Nitrate assimilation is tightly controlled by phosphorylation- or dephosphorylation-mediated post-translational regulation [32]. *PP2C* participates in the regulation of ammonium transporters (AMTs). Under moderate ammonium supply or high nitrogen demand, *PP2C* regulates ammonia transport and uptake by activating *CIPK23*. The protein phosphatase family, consisting of 150 members in Arabidopsis and 130 members in rice, plays important roles in nitrogen uptake and assimilation pathways [33,34]. For instance, protein phosphatase *PP2C9* is closely related to improving nitrogen uptake and assimilation in rice [35]. One recent study demonstrated that CEPD-induced phosphatase (*CEPH*), a member of the *PP2C* gene family, acts as a target gene of nitrogen long-distance signaling. *CEPH* acts as a key enzyme that regulates nitrate uptake in plants at the post-translational level by dephosphorylating NRT2.1 in response to the complex nitrogen environment [18]. Given the significance of maize as a major food crop, it is crucial to investigate whether maize possesses similar mechanisms to cope with the heterogeneity of soil nitrogen environments. Although previous studies identified *ZmPP2C* genes in maize and analyzed their evolution and expansion [36,37], there is a research gap concerning the latest version of the reference genome and nitrogen-related functions, especially in the context of heterogeneous nitrate availability. In this study, we present a comprehensive analysis of the maize *PP2C* gene family. Through evolutionary analysis with *Arabidopsis thaliana*, we identify homologous genes and examine their expression patterns under different nitrogen treatments, as well as their responses to long-distance nitrogen signals. In this way, we aim to gain initial insights into the long-distance nitrogen-signaling mechanisms in maize, setting the stage for future in-depth investigations in this field.

## 2. Results

### 2.1. Identification of PP2C Members in the Maize Genome

Initially, a total of 229 nonredundant putative candidate genes were identified through a hidden Markov model (HMM) search. Subsequently, a BLASTP search was performed using the AtPP2Cs proteins as queries against the *Zea mays* B73 v5 genome, resulting in 893 candidate genes. By intersecting the results of the HMM and BLASTP searches, 198 candidate genes were obtained. To ensure the presence of the PP2C domain, all encoded PP2C proteins were checked manually using the NCBI Batch to CDD tool. As a result, 95 proteins lacking the PP2C domain were deleted, yielding a final set of 103 PP2Cs with intact PP2C domains.

For simplicity, the identified 103 *PP2C* genes were designated as *ZmPP2C1* to *ZmPP2C103* according to chromosomal position (Appendix A). These *PP2C* genes encode proteins with carrying lengths ranging from 106 to 1084 amino acids (aa). Analysis of the PP2C protein sequences revealed significant diversity in their characteristics. The predicted molecular weight of the 103 PP2C proteins ranged from 11.07 to 120.5 kDa. The isoelectric point varied from 4.07 to 10.34, and the instability index ranged from 26.24 to 63.69. More detailed information on these data is shown in Appendix A. The predicted subcellular localization results indicated that most PP2C proteins were distributed across various organelles, such as the chloroplast, nuclear region, mitochondria, and cytoplasm.

### 2.2. Evolutionary and Comparative Analysis of the PP2C Family in Four Species

To explore the evolutionary relationship and potential functional characteristics of the *PP2C* family members in maize, rice, Arabidopsis, and tomato, a comparative evolutionary tree was constructed, aligning 353 full-length protein sequences from the four species. Based on the branch position of the *ZmPP2C* gene family members, we divided the 103 *ZmPP2Cs* into 11 subgroups, designated as A-J (Figure 1, Table 1). The A and H groups encompass the largest number of *ZmPP2C* genes, with 16 genes each, while the G subgroup has only 2 genes. Notably, we observed that 13 *ZmPP2C* genes belong to the same subgroup as *AtCEPH* in Arabidopsis. However, it is worth mentioning that there are no genes in maize that exhibit homology to *AtCEPH*.

To investigate the evolutionary relationships and replication events within the *PP2C* gene family, we analyzed the genome-wide collinear relationship between maize, rice, and Arabidopsis. This analysis revealed 98 orthologous gene pairs between maize and rice and 20 orthologous gene pairs between maize and Arabidopsis. (Figure 2).

To gain insights into the selective pressures acting on the duplicated *PP2C* genes, we calculated the Ka/Ks ratio. Generally, a Ka/Ks ratio > 1 means adaptive evolution with positive selection, a ratio of 1 indicates neutral selection, while a ratio < 1 means negative or purifying selection [38]. In our study, all the Ka/Ks ratios for the 50 segmentally duplicated gene pairs were less than 1 (Appendix A), indicating that the *ZmPP2C* gene family has undergone purifying selection and has maintained a high level of conservation throughout its evolutionary history.

### 2.3. Gene Structure, Conserved Motif, and Evolutionary Analysis of ZmPP2Cs

To gain deeper insight into the relationship between the structure and function of ZmPP2C proteins, we analyzed the gene structure and conserved motifs and domain based on evolutionary relationships. The exon–intron structures were analyzed to reveal the gene structural features (Figure 3). Notably, the introns and exons of genes in the same branch and subclade are highly conserved. The number of exons ranged from 1 to 17 among the *ZmPP2C* genes, with most genes containing 4 exons, 4 genes only containing 1 exon, and 2 genes containing no introns.

To analyze the presence of conserved motifs in different ZmPP2C proteins and gain a better understanding of the similarities and differences, we used the MEME tool to identify 20 distinct conserved motifs, designated as motifs 1–20 (Table 2). The composition patterns of motifs tend to be consistent with the evolutionary tree results (Figure 4A). The motifs within the same clade are highly conserved, while the motif composition diversity varies among different clades (Figure 4B). Motifs 1, 2, 3, and 4 are ubiquitously presented in *ZmPP2Cs*, indicating that these motifs may share conserved positions and functions. However, significant differences in motif were observed between branches, suggesting the diversity of the maize *PP2C* gene family in the evolutionary process. In addition to the PP2Cc domain, *ZmPP2C18* has a PRK14951 superfamily, PKc_like superfamily, and CAP_ED domains. This suggests that this gene may have other functions beyond being a protein phosphatase.

### 2.4. Chromosomal Localization and Gene Duplication of ZmPP2C Gene Family

We mapped *ZmPP2Cs* onto the maize genome based on the physical location information. Chromosome localization analysis showed that the distribution of chromosome localization was uneven (Appendix A). Generally, these 103 *ZmPP2Cs* were unevenly distributed on the 10 chromosomes. Among them, chromosome 2 contained the largest number of *ZmPP2Cs*, with 15 genes (14.56%), while chromosome 3 had only 5 *ZmPP2Cs* (4.85%) (Appendix A). This uneven distribution of *ZmPP2Cs* on the chromosomes reflected the diversity and complexity of the *ZmPP2C* gene family.

We define genes with a distance of less than 200 kb as tandem duplicates, while others are considered segmental copies. Among the *ZmPP2C* genes, three gene pairs were involved in gene duplication events (Appendix A). Additionally, the three gene pairs are recognized as segmental duplication according to the above classification method, indicating that tandem duplication events might not have taken part in amplifying the *ZmPP2C* gene family.

### 2.5. Cis-Element Analysis in the Promoters of ZmPP2Cs

To investigate the potential roles of *ZmPP2Cs* in abiotic stresses, we analyzed the cis-elements of promoter regions (2.0 kb upstream of the ATG start codon) of all 103 genes using the PlantCARE online tool. A total of 26 types of cis-acting elements involved in stress and phytohormone responses were identified, including MeJA responsiveness, defense and stress response, auxin, ABA, and gibberellin (Appendix A). Among these elements, the light-responsive element and the ABA-responsive element were the most abundant and present in the promoters of all 103 *ZmPP2C* genes (Appendix A). Auxin-responsive elements were also frequently present in *ZmPP2C* gene-promoter regions— they were found in 75 promoters. Elements related to defense and stress responsiveness were detected in only 31 *ZmPP2C* gene promoters. Additionally, low-temperature and drought-stress elements were detected in the promoters of 53 and 58 *ZmPP2C* genes, respectively. Many other functional elements were also identified and listed in Appendix A. These findings suggest that *ZmPP2C* genes may play crucial roles in biological and abiotic stress.

### 2.6. Protein–Protein Interaction Network of ZmPP2Cs

Protein–protein interaction networks (PPIs) allow for elucidating relationships between proteins, which may be correlated through specific functional interactions [38]. To gain a better understanding of the relationship among *ZmPP2C* genes at the protein level, we constructed a protein–protein interaction (PPI) network for the 103 ZmPP2C proteins. We removed outliers in the network graph, which finally resulted in three clusters (Figure 5). Based on the analysis of the PPI networks, we found that ZmPP2C45 and ZmPP2C95 exhibited stronger interactions with other proteins within the network. This suggests that these two ZmPP2C proteins may play major roles in the *PP2C* family that we identified.

### 2.7. Expression Profiles of ZmPP2C Genes in Different Maize Tissues

In order to further explore the role of *ZmPP2C* genes, we analyzed their expression patterns in 10 different maize tissues, including internode, ear primordium, embryo, mature leaf, primary root, root cortex, root elongation zone, root meristem zone, mature pollen, and silk (Figure 6). Our analysis revealed that most *ZmPP2C* genes are expressed in all tissues, with some exceptions. For example, *ZmPP2C42, ZmPP2C47,* and *ZmPP2C13* were expressed at higher levels throughout all tissues, and we suspected that these genes played key roles in maize growth and development. *ZmPP2C42* and *ZmPP2C47* showed the highest expression in mature pollen, suggesting their potential roles in mature pollen. Interestingly, some genes, such as *ZmPP2C15* and *ZmPP2C95*, are specifically highly expressed in the root cortex. In addition, *ZmPP2C59* was found to be expressed in the primary root and root cortex. These findings suggest that these genes may have specialized functions in these specific tissues.

### 2.8. Expression of ZmPP2C Genes under Different Nitrogen Concentration Stress

To further investigate the expression patterns of *ZmPP2C* genes under nitrogen stresses, with the information from the public database, we selected 10 genes from the 103 *ZmPP2C* genes that showed potential responsiveness to nitrogen long-distance signals in roots. We examined the expression of these ten *ZmPP2C* genes in response to high- and low-nitrogen conditions via qRT-PCR analysis. The qRT-PCR results showed that the expression of *ZmPP2C37, ZmPP2C41, ZmPP2C50, ZmPP2C75,* and *ZmPP2C85* was significantly upregulated in response to nitrogen deficiency treatment (0 mM NO_3_^−^) and peaked at 24 h after treatment. Notably, *ZmPP2C37* and *ZmPP2C50* showed significant differences at all three time points after nitrogen deficiency treatment compared with the initial time point (0 h) (Figure 7A). Under high-nitrogen treatment (15 mM NO_3_^−^), the expressions of *ZmPP2C41, ZmPP2C75,* and *ZmPP2C85* were significantly inhibited at 48 h compared with the control. *ZmPP2C54* and *ZmPP2C81* were not differentially expressed under high-nitrogen treatment (Figure 7B). These results suggest that the genes of the *ZmPP2C* family have different response patterns to nitrogen stress and provide insight into the potential roles of *ZmPP2C* genes in regulating nitrogen stress responses in maize.

### 2.9. Differential Gene Expression of the ZmPP2Cs under Heterogeneous Nitrogen Stress

We further investigated how these eight *ZmPP2C* genes respond to heterogeneous nitrogen conditions via the split-root and shoot removal experiments (Appendix A). Results showed that the expression levels of *ZmPP2C54* and *ZmPP2C85* were downregulated under heterogeneous nitrogen treatment, which indicated that these genes were negatively regulated by the nitrogen-deficiency signal derived from the other side of the root system in response to the heterogeneous nitrate environment (Figure 8). However, when the shoot was removed, this negative regulation was eliminated (Figure 8). These results highlight the dynamic and complex regulation of *ZmPP2C* genes under heterogeneous nitrogen conditions, indicating that the *ZmPP2C* family plays a crucial role in the response to long-distance nitrogen signals.

### 2.10. Subcellular Localization

The subcellular localization of the ZmPP2C85 protein was predicted in the nucleus with the online tool Plant-mPLoc (Appendix A). In order to further verify the results, the cDNA of *ZmPP2C85* without the stop codon was fused to the N-terminus of the eGFP reporter gene and inserted into an expression vector under the control of the CaMV 35S promoter. The constructed fusion protein was then transiently transformed into tobacco cells. Under a laser confocal microscope, we found that ZmPP2C85-eGFP was localized in the nucleus (Figure 9), consistent with the prediction results. This observation suggests that ZmPP2C may play a key role in the nucleus.

## 3. Discussion

PP2C-type phosphatases form a major class of phosphatases in plants and play a crucial role as regulators of signal transduction pathways and are involved in development [39]. To date, *PP2C* genes have been identified in various plant species, such as *Arabidopsis thaliana* [40], *Oryza sativa* [26,40], *Brachypodium distachyon* [21], *Populus euphratica* [41], *Glycine max* and *Sorghum bicolor* [42], *Brassica rapa* [43], *Glycyrrhiza uralensis Fisch* [44], *Brassica juncea* var. *tumida* [45], and *Zea mays* [36,37]. In a previous report, Wei and Pan characterized the protein phosphatase superfamily in maize, including 104 members in the *ZmPP2C* family, based on the maize B73 RefGen_v2 annotation [36]. Fan et al. analyzed the evolution and expansion of 97 *ZmPP2C* family members based on the maize B73 RefGen_v3 annotation [37]. Due to the substantial changes in genome assembly and annotation among the newly released Zm-B73-GRAMENE-NAM-5.0 and previous versions, there is a research gap concerning the latest genome version and the nitrogen-related functions of the *PP2C* family, especially in the context of heterogeneous nitrate availability. Motivated by these research gaps, we conducted this study to explore these questions and provide new insights into the nitrogen-related functions of *ZmPP2C* genes.

In this study, a total of 103 *ZmPP2C* genes were identified (Appendix A). We also studied the distribution of this family on chromosomes, which provided valuable information about the genomic organization of the *ZmPP2C* gene family and their arrangement on specific chromosomes. According to evolutionary analysis (Figure 1), the *PP2C* genes of maize, rice, Arabidopsis, and tomato were further categorized into 11 subgroups, which may have similar functions in each branch. Gene duplication is one of the staple driving forces of biological evolution [38], and it may contribute to the diversity of *ZmPP2C*. In this study, 50 *ZmPP2C* tandem repeat gene pairs were identified. Ka/Ks of these tandem repeats were calculated. The result showed that Ka/Ks for 50 pairs of duplicated *ZmPP2C* genes was <1, suggesting that all duplicated *ZmPP2C* genes have mainly evolved from the purifying selection. This conclusion was mutually corroborated by the fact that the members of the *ZmPP2C* gene family were conserved.

We executed the MEME program to predict conserved motifs by analyzing the sequence features of *ZmPP2C*. Motifs 1, 2, and 3 are ubiquitously present in *ZmPP2C*, indicating their highly conserved domain (Figure 4). Different subgroups contain their own specific motifs, which may lead to the functional divergence of each subgroup. Cis-elements play important roles in regulating gene expression and are particularly important in revealing their potential functions [46]. We analyzed the promoters of *ZmPP2C* genes and found that they contained a large number of cis-acting elements, including light-, methyl jasmonate-, defense- and stress-, gibberellin-, anaerobic-, wound-, and ABA-responsive elements (Appendix A). Previous studies have shown that GCN4 motifs are involved in the response to nitrogen stress [47]. In our study, there are three *ZmPP2C* genes, including *ZmPP2C19, ZmPP2C37*, and *ZmPP2C71*, containing GCN4 motifs, which suggests that these genes may be involved in the nitrogen-stress response.

A previous study has shown that *AtCEPH* is involved in the long-distance nitrogen signaling pathway in Arabidopsis [18]. To investigate whether *ZmPP2Cs* have similar functions to *AtCEPH*, we conducted in-depth analyses of the maize genes in the evolutionary branch of *AtCEPH*. We constructed an evolutionary tree and further analyzed their motifs and domains (Appendix A). However, evolutionary analysis did not find an orthologous gene for *AtCEPH* in the maize genome. Nevertheless, we identified *ZmPP2C41, ZmPP2C50,* and *ZmPP2C75* as the closest paralogous genes. *ZmPP2C41* and *ZmPP2C50* have similar motifs to *AtCEPH*, which indicates that these genes may have similar functions to *AtCEPH* and should receive in-depth analysis. Probing further, we found that *ZmPP2C54* and *ZmPP2C69* may be involved in long-distance nitrogen signaling [7]. After searching in the plant public RNA-seq database [48], we identified another candidate gene *(ZmPP2C81*) that might be involved in nitrogen responses. Therefore, to further understand whether these genes respond to nitrogen stress, we performed qRT-PCR on the ten candidate genes and found that the expression of *ZmPP2C37, ZmPP2C41, ZmPP2C50, ZmPP2C54, ZmPP2C69, ZmPP2C75,* and *ZmPP2C85* was perturbed by nitrogen stress, with the expression of *ZmPP2C37*, *ZmPP2C41, ZmPP2C75,* and *ZmPP2C85* most significantly upregulated under nitrogen deficiency for 24 h (Figure 7A). These early responses of *ZmPP2C37*, *ZmPP2C41, ZmPP2C75*, and *ZmPP2C85* suggest their key regulatory role under nitrogen deficiency.

Split-root experiments and shoot removal experiments are classic physiological methods in the verification of long-distance signaling [49,50]. Our previous studies have established that the shoot plays a central role in long-distance signaling [51,52]. In this study, we found that *ZmPP2C41*, *ZmPP2C50*, and *ZmPP2C75,* which are the closest paralog genes relative to *AtCEPH*, were not affected by heterogeneous nitrogen treatment, which implied a different long-distance mechanism in maize compared with Arabidopsis (Figure 8). Nevertheless, under the treatment of a heterogeneous nitrate environment, we found that the expression of *ZmPP2C85* and *ZmPP2C54* was downregulated. Further validation through the shoot removal treatment revealed that the expression of *ZmPP2C54* and *ZmPP2C85* did not respond to the nitrogen deficiency signal of heterogeneous nitrate treatment, indicating that the genes were regulated by long-distance nitrogen signals (Figure 8).

The *ZmPP2C85* is an orthologous gene of *SbPP2C9* in sorghum (Appendix A). *SbPP2C9* was found to be closely related to enhanced nitrogen use efficiency by promoting nitrogen uptake and assimilation [35]. Since maize and sorghum are highly evolutionary related, we hypothesized that *ZmPP2C85* may play a similar role as *SbPP2C9* in regulating nitrogen uptake and assimilation. Genes perform corresponding functions by being translated into proteins, and the location of translated proteins determines their functions. Therefore, in order to better explore its function, we used tobacco transient expression for subcellular localization, and the results showed that the ZmPP2C85-eGFP was located in the nucleus, which was consistent with the predicted results. Meanwhile, the *PP2C* gene was also reported to localize in the nucleus in *Arabidopsis thaliana* [53], *Pyrus bretschneideri* [54], and *woodland strawberry* [55], which further suggested that the gene might function in the nucleus. The localization of ZmPP2C85-eGFP is different from the AtCEPH-GFP, which was localized in both the nucleus and cytoplasm of root epidermal cells [18], although these differences might at least partly be attributed to the differences in the experimental system. These results suggest further investigations are warranted to unravel the precise role of *ZmPP2C85* in plant development, long-distance signaling pathways, and stress responses.

## 4. Conclusions

In this report, we identified 103 *PP2C* genes in maize. This study provides important information on the identification, classification, promoter analysis, evolutionary relationships, and expression patterns of the *ZmPP2C* gene family in different tissues. Meanwhile, we found that the genes of this family responded to different nitrogen stresses and further in-depth analysis found that some responded to nitrogen long-distance signals. Our subcellular localization analysis of the important candidate gene *ZmPP2C85* indicated that the gene may function in the nucleus. Moreover, our findings contribute to our understanding of the roles of *ZmPP2C* genes in nitrogen regulation and provide a foundation for future research on improving nitrogen use efficiency in maize.

## 5. Materials and Methods

### 5.1. Database Searches and Genome-Wide Identification of PP2C Family Members in Maize

To identify all members of *PP2Cs* in the maize genome, we employed two strategies. First, we used all PP2C protein sequences in Arabidopsis as a reference and performed local blasts with an E-value of 1 × 10^−5^ via the TBtools software (Version 1.098721) [56]. Additionally, the hidden Markov model (HMM) profile of the PP2C domain (PF00481) was downloaded from the Pfam database (http://pfam.xfam.org/, accessed on 6 May 2022) and searched against all protein sequences of the maize genome. Next, we selected the genes that intersected the two sets of results for further analysis. Then, the NCBI Conserved Domain database (https://www.ncbi.nlm.nih.gov/Structure/cdd/wrpsb.cgi, accessed on 6 May 2022) was utilized for the domain prediction, and the genes without the PP2C domain were manually removed, resulting in a final total of 103 *PP2C* members. The amino acid sequences, genome sequences, and GTF/GFF files of the maize v5 genome were downloaded from the Ensembl Plants website (https://plants.ensembl.org/index.html, accessed on 6 May 2022).

### 5.2. Analysis of Maize PP2C Protein Feature

The Compute pI/MW tool available on the ExPASy server (http://web.expasy.org/compute, accessed on 6 May 2022) was used to calculate various protein features of ZmPP2C proteins. These included the molecular weight (MW), the theoretical isoelectric point (pI), the instability index, the aliphatic index, and the grand average of hydropathicity (GRAVY). Subcellular localization of plant proteins was predicted via the machine learning algorithms embedded in Plant-mPLoc (http://www.csbio.sjtu.edu.cn/bioinf/plant-multi/, accessed on 6 May 2022).

### 5.3. Evolutionary Tree, Gene Structure, and Conserved Motifs

The protein sequences of *PP2C* genes in maize, rice, Arabidopsis, and tomato were aligned using the Clustalw algorithm in MEGA X (64-bit) for evolutionary analysis with default parameters. The Dayhoff model was selected as the best model based on the lowest BIC scores. Additionally, an evolutionary tree was constructed using the neighbor-joining (NJ) method, and bootstrap analysis with 100 replicates was performed to assess the support for the tree topology. Finally, FigTreev1.4.4 was used to visualize and refine the evolutionary tree. TBtools software was used to analyze the introns and exons in conjunction with the Chr GTF files.

Conserved motifs were identified using the MEME program (http://meme-suite.org/, accessed on 6 May 2022). Parameter settings were optimized with a maximum number of motifs set as 15, while other parameters were left as default. Domain information was downloaded and analyzed in NCBI Batch CD Search with default parameters (https://www.ncbi.nlm.nih.gov/Structure/bwrpsb/bwrpsb.cgi, accessed on 6 May 2022). All analyses were performed in TBtools.

The protein sequence of SbPP2C was downloaded from the Ensembl Plants website. The “Quick find best homology” function of the TBtools software was used to construct the evolutionary tree with default settings.

### 5.4. Chromosomal Location and Synteny Correlation Analysis

The physical locations of *PP2C* genes on maize chromosomes were extracted from the maize genomic GFF file with TBtools. We used TBtools to extract the information for the 10 chromosomes and organized them into chromosome density files. The physical location map was then constructed with modifications to color parameters for improved visualization. For synteny gene analysis, the relationships between homologous genes of maize, rice, and Arabidopsis were examined and visualized using TBtools.

### 5.5. Identification of Cis-Regulatory Elements in Promoters of ZmPP2Cs

To identify potential cis-regulatory elements in the promoter sequences of *ZmPP2C* genes, the 2000 bp sequences of each *ZmPP2C* gene upstream of the ATG start codon were extracted from the maize genome. The PlantCARE program (http://bioinformatics.psb.ugent.be/webtools/plantcare/html/, accessed on 6 April 2022) was employed to analyze and identify cis-acting elements within the *ZmPP2C* gene promoters. The resulting irrelevant or redundant information was deleted manually. TBtools was used to visualize cis-regulatory elements.

### 5.6. Protein–Protein Network Interaction Analysis of ZmPP2C Genes

We used the String web-based program (version 11.5) (http://www.string-db.org/, accessed on 25 May 2022) to construct a protein interaction network of the *ZmPP2C* genes using the default parameters. Then, the Cytoscape tool (Version3.9.1) was used to improve visualization.

### 5.7. Expression Analysis of ZmPP2C Genes between Different Tissues

To analyze the expression of *ZmPP2C* genes in different tissues at different growth stages, the data were downloaded from publicly available transcriptome data from MaizeGDB (http://qteller.maizegdb.org/rna_data_sources.php, accessed on 10 June 2022). The expression values were transformed by log2 (RPKM+1), and heatmaps were visualized with TBtools software.

### 5.8. Plant Growth and Treatments

Seeds of B73 were surface sterilized in 10% H_2_O_2_ solution for 30 min, followed by washing in distilled water and with 2% H_2_O_2_ solution in the dark. Sterilized seeds were grown on a growth matrix until the 3-leaf stage. At this stage, the endosperm was carefully removed from the seedlings. Ca (NO_3_)_2_ was supplied as an N source. CaCl_2_ was an additional calcium source under nitrogen deficiency treatment. The nutrient solution pH was adjusted to 5.8–6.0. The plants were grown under controlled conditions with a 14 h light period at 28 °C and a 10 h dark period at 22 °C. The humidity was maintained at 50%, and the light intensity was set at 500 µmol m^−2^ s^−1^.

In the nitrate treatment experiment, we conducted two treatments of nitrogen deficiency (0 mM) and high nitrogen (15 mM), respectively, for 0 h, 12 h, 24 h, and 48 h, and each treatment was replicated three times, with three plants per replicate.

### 5.9. Split-Root Experiments

The uniform seedlings at the 3-leaf stage were transferred to a hydroponics system with Hoagland solution. Once the plants reached the 5-leaf stage, a split-root treatment was performed. The split-root system of a plant was separated into left (L) and right (R) parts that were exposed to different nutrient conditions. Roots were subjected to four different treatments: (1) plants received N (Sp. + N) on one side and nitrogen deficiency (Sp. − N) on the other (HeteroHN); (2) plants received N on both sides of the root system (HomoHN); (3) plants received N on both sides of the root system, and shooting was removed (Remove-HomoHN); (4) plants received N (Sp. + N) on one side and nitrogen deficiency (Sp. − N) on the other, and shooting was removed (Remove-HeteroHN) (Appendix A). Each treatment had three replicates, three plants per replicate.

### 5.10. RNA Extraction and qRT-PCR

Total RNA was isolated from maize root with Trizol according to the manufacturer’s protocol (Aidlab, Beijing, China). The cDNA was synthesized using the HiScript III RT SuperMix for qRT -PCR (+gDNA wiper) kit (Vazyme, Nanjing, China) according to the manufacturer’s protocol. The primers were designed using the IDT Primer Quest Tool (version 2.2.3) (https://sg.idtdna.com/Primerquest/Home/Index, accessed on 10 March 2023). Real-time quantitative PCR (qRT-PCR) was conducted on a CFX96 Touch™ Real-Time PCR detection system (BIO-RAD, Hercules, CA, USA) using ChamQ SYBR qPCR Master Mix (Vazyme, Nanjing, China). A 10 µL reaction volume for each sample containing 5 µL of 2 × ChamQ SYBR qPCR Master Mix (Vazyme, Nanjing, China), 0.2 µL of each primer, 1 µL of diluted cDNA product, and 3.6 µL of ddH_2_O. The qRT-PCR reaction procedures were as follows: 3 min at 95 °C for DNA polymerase activation, denaturation, and anneal/extension at 95 °C for 10 s and 60 °C for 30 s, respectively, for a total of 40 cycles. A melting curve was used to evaluate the specificity of amplification. The maize *Actin* gene (NC_024466) was used as the internal control for normalization. Each quantitative real-time PCR (qRT-PCR) procedure was performed in triplicate. The relative mRNA expression level for each gene was calculated using the 2^−ΔΔCT^ method. The statistical significance among three biological replicates was tested based on Student’s *t*-tests. *p* ≤ 0.05 was regarded as statistically significant. All primers are listed in Appendix A.

### 5.11. Subcellular Localization Analysis

The full-length CDS sequence of *ZmPP2C85* without stop codon was fused to the N-terminus of the eGFP sequence with expression driven by the CaMV 35S promoter. The pcambia2300-eGFP-ZmPP2C85 fused proteins were transiently transformed into *Nicotiana Benthamiana* leaves. The eGFP fluorescence was observed using a confocal microscope (Zeiss LSM 800, Carl Zeiss Microimaging Inc., New York, NY, USA).

## Figures and Tables

**Figure 1 plants-12-03153-f001:**
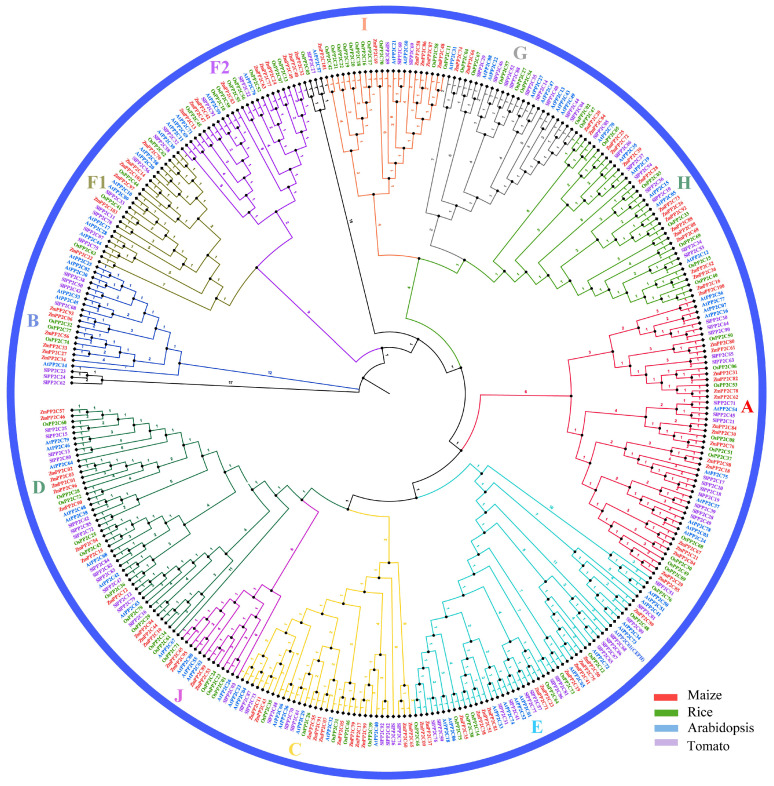
Evolutionary tree of PP2C proteins among maize, rice, Arabidopsis, and tomato. Red, green, blue, and purple schemes represent maize, rice, Arabidopsis, and tomato, respectively. The evolutionary tree classifies all the *PP2Cs* into eleven (A–J) subgroups; the eleven subgroups are represented by different colors.

**Figure 2 plants-12-03153-f002:**
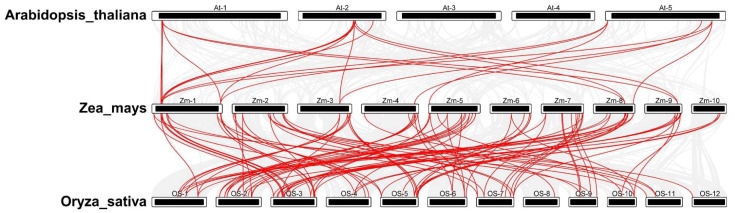
Synteny analysis of *ZmPP2C* genes with the genomes of Arabidopsis and rice. There are 20 orthologous gene pairs between maize and Arabidopsis and 98 orthologous gene pairs between maize and rice.

**Figure 3 plants-12-03153-f003:**
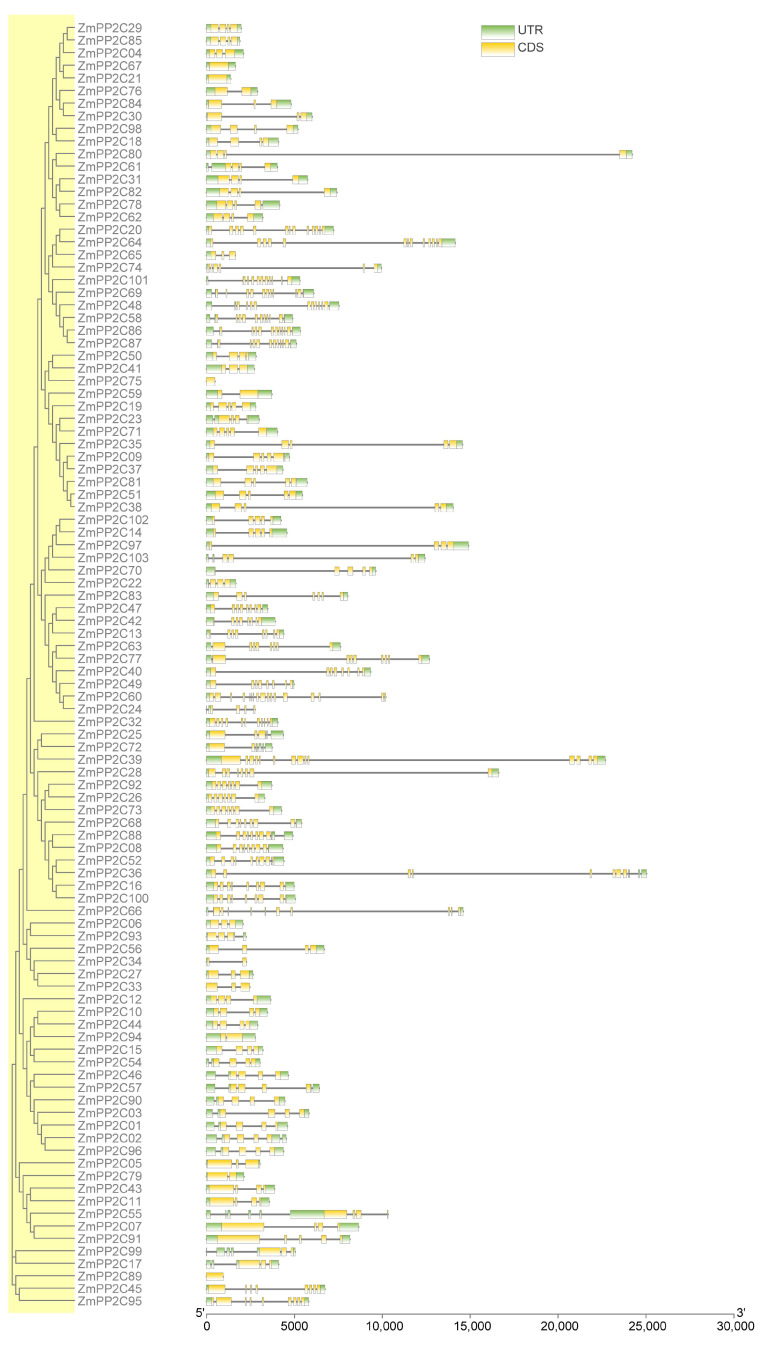
Gene structure of *ZmPP2C* genes. Black lines indicate introns, and CDS and untranslated regions (UTR) are indicated by yellow and green boxes, respectively. The ruler at the bottom is used to estimate their length.

**Figure 4 plants-12-03153-f004:**
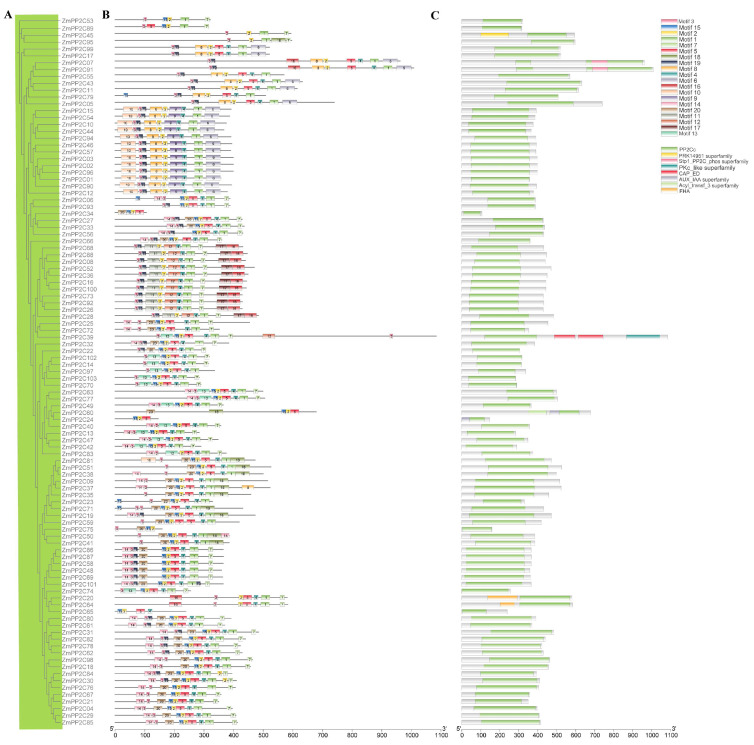
Evolutionary tree, motif, and domain of *ZmPP2C* genes. (**A**) The evolutionary tree was constructed in MEGA X software (64 bit) using the neighbor-joining (NJ) method and 1000 bootstrap tests. (**B**) Conserved motifs were identified using the website of MEME and TBtools software (Version 1.098721); different colors indicate different motifs. (**C**) Domains were predicted through the website of NCBI Batch to CDD and plotted in TBtools software (Version 1.098721). The length of each protein can be estimated using the scale at the bottom.

**Figure 5 plants-12-03153-f005:**
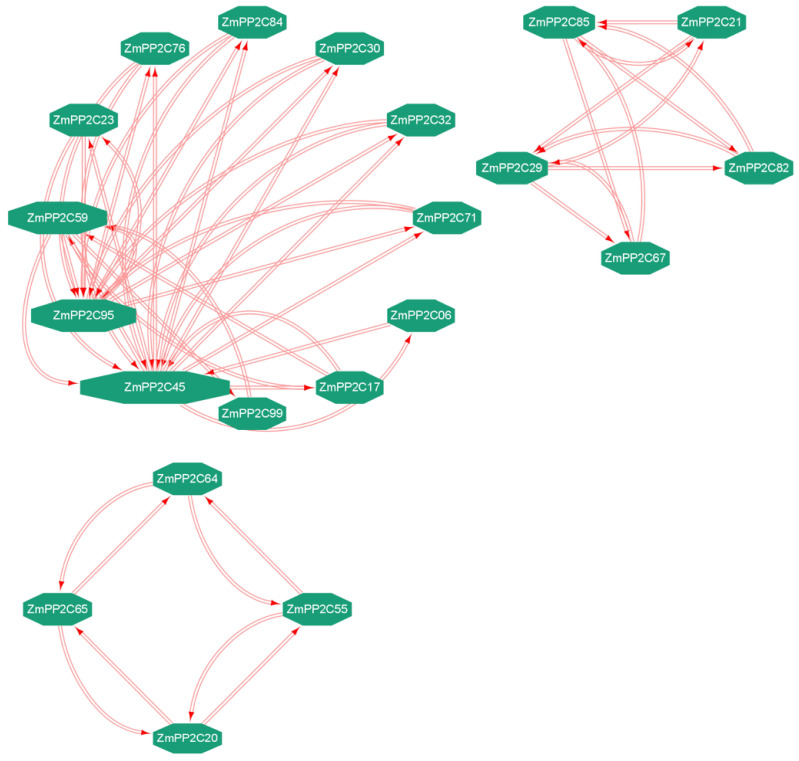
Protein–protein interactions of ZmPP2C proteins predicted using the STRING tool.

**Figure 6 plants-12-03153-f006:**
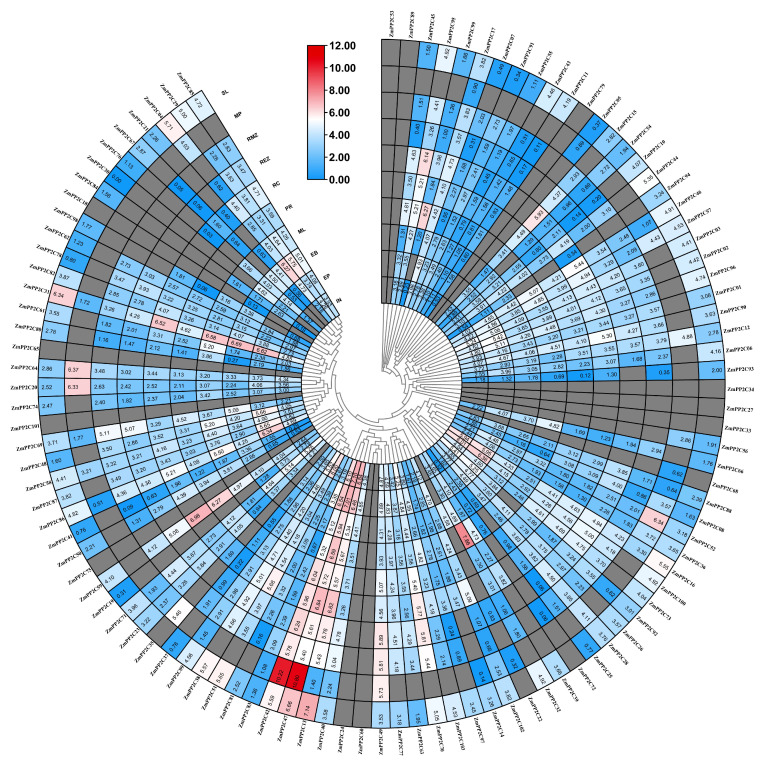
Heatmap of expression profiles of *ZmPP2C* gene family members in different tissues. Ten tissues from different developmental stages were investigated, including internode (IN), ear primordium, embryo, mature leaf, primary root, root cortex, root elongation zone, root meristem zone, mature pollen, and silk. The expression values are shown as log2 of the RPKM+1 values. Gray color represents “NA”.

**Figure 7 plants-12-03153-f007:**
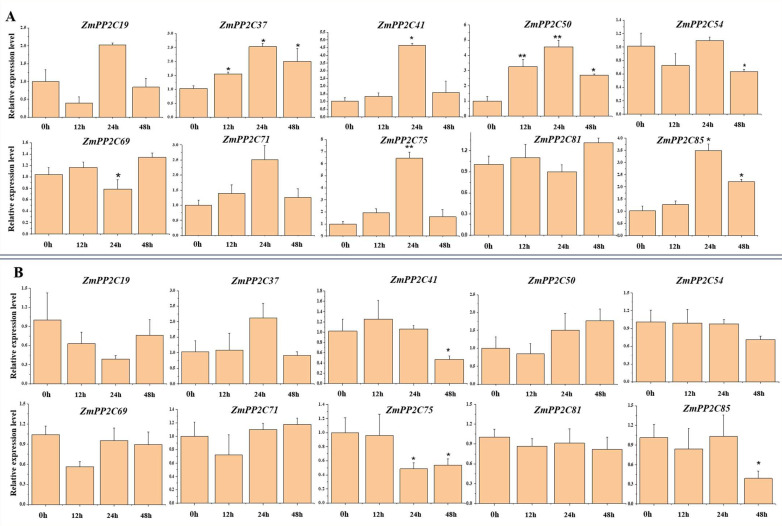
Expression of *ZmPP2C* genes in response to different nitrogen treatments: (**A**) expression changes of 6 *ZmPP2C* genes at 0 h, 12 h, 24 h, and 48 h under nitrogen-deficiency stress (0 mM NO_3_^−^); (**B**) expression changes of the 6 genes at 0 h, 12 h, 24 h, and 48 h under high-nitrogen stress (15 mM NO_3_^−^). Data were statistically analyzed using the *t*-test, * means *p* < 0.05; ** means *p* < 0.01.

**Figure 8 plants-12-03153-f008:**
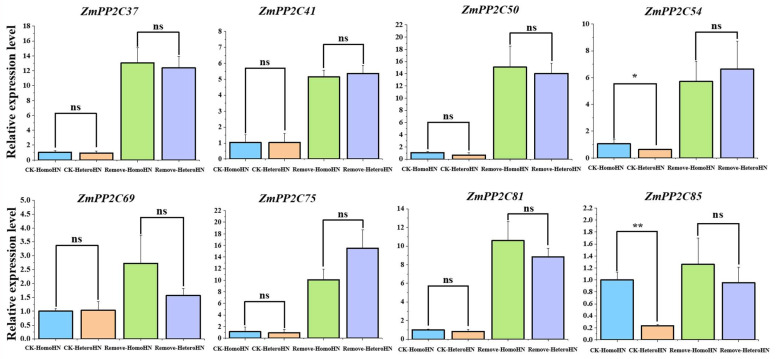
Expression of *ZmPP2C* genes in response to heterogeneous nitrogen stress. Blue represents the expression level under homogeneous nitrate treatment conditions (CK-HomoHN); orange represents the expression level under heterogeneous nitrate treatment conditions (CK-HeteroHN); green represents the expression level under homogeneous nitrate treatment conditions after removal of shoots (Remove-HomoHN); and purple represents the expression level under heterogeneous nitrate treatment conditions after removal of shoots (Remove-HeteroHN). Data were statistically analyzed using the *t*-test. ns means not significant, * means *p* < 0.05; ** means *p* < 0.01.

**Figure 9 plants-12-03153-f009:**
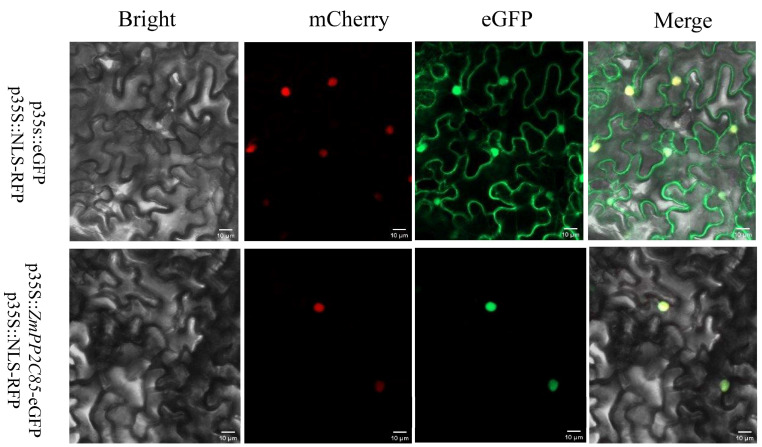
Subcellular localization of ZmPP2C85-eGFP fusion protein in *Nicotiana benthamiana* leaves. p35S::eGFP was used as a control. Green, eGFP signal. Red, a nuclear maker signal. Bright-field illumination is shown in Column 1, mCherry signal is shown in Column 2, eGFP signal in Column 3, and a merge of all signal patterns in Column 4. The scale bar indicates 10 μm.

**Table 1 plants-12-03153-t001:** The distribution of *PP2C* gene numbers in Arabidopsis, rice, maize, and tomato.

Subgroup of *PP2C* Genes	Numbers of *AtPP2Cs*	Numbers of *OsPP2Cs*	Numbers of *ZmPP2Cs*	Numbers of *SlPP2Cs*
A	10	10	16	15
B	6	3	6	4
C	7	5	11	8
D	9	10	13	14
E	14	9	13	15
F1	8	4	6	7
F2	5	8	9	3
G	8	7	2	9
H	6	6	16	8
I	2	11	6	4
K	4	4	4	1
Others	1	1	1	4
Total	80	78	103	92

*AtPP2Cs*: Arabidopsis thaliana *PP2Cs*; *OsPP2Cs*: Oryza sativa *PP2Cs*; *ZmPP2Cs*: Zea mays *PP2Cs*; *SlPP2Cs*: Solanum lycopersicum *PP2Cs*.

**Table 2 plants-12-03153-t002:** Conserved motifs in the amino acid sequences of *ZmPP2C*.

Motif	Width	Multilevel Consensus Sequence
1	29	EPEVTVVEJSPDDEFLILASDGLWDVLSN
2	15	LYVANVGDSRAVLSR
3	11	SFFGVFDGHGG
4	15	GGLAVSRAIGDRYLK
5	29	GGKAVQLSVDHKPBRPDERERIEAAGGRV
6	50	PRGGIARRLVKAALQEAAKKREMRYSDLKKIDRGVRRHFHDDITVVVVFL
7	15	RGSKDBITVIVVDLK
8	41	DAJRKAFEATEEGFLSLVEKEWSLKPZJASVGSCCLVGVIC
9	41	VAEQLSAEHNASYEEVRQELQSSHPDDPHIVVLKHNVWRVK
10	50	HRGKGSDAAGRQDGLLWYKDLGQHVAGEFSMAVVQANQLLEDQSQVESGP
11	41	EDWLAALPRALVAGFVKTDKDFQTKAETSGTTVTFVIIDGW
12	41	SADHRLDANEEEVERVTASGGEVGRLNVVGGAEIGPLRCWP
13	41	AAEYLKEHLFENJLKHPEFITDTKLAISETYQKTDSEFLEA
14	27	FGLSSVRGRRAEMEDAVAVRPDFDDGT
15	15	TSGSTAVTAVVVGGH
16	41	KPGZGISVHEGSSKSSKLRPWGGPFLCSSCQEKKEAMEGKR
17	37	EEIFEEGSAMLSRRLNSNYPVRNMFKLFRCAICQVDL
18	48	KEVVDIVSSAPSRASAARALVESAVRAWRTKYPTSKVDDCAVVCLFLB
19	15	PEAADFLREHLYNNV
20	29	AEMAAAWREAFERAFARMDEELKGQAGVD

## Data Availability

Data sets supporting the conclusions and descriptions can be found in the manuscript and annexes. The complete data set used and analyzed in this study is available from the corresponding author upon reasonable request.

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
