# Peer review of "Genome-Wide Identification and Characterization of the PP2C Family from Zea mays and Its Role in Long-Distance Signaling"

_plants, 2023, doi:10.3390/plants12173153_

Round 1
Reviewer 1 Report
The present study focuses on the PP2C family in Zea mays. Authors identified 103 ZmPP2C genes in maize genome and estimated their expression levels based on available transcriptomic data. Moreover, for six ZmPP2C genes, the expression levels were analyzed by qPCR in the roots under different nitrogen status. Overall, the authors performed comprehensive analysis on identification and characterization of ZmPP2C genes. However, the results on regulation of ZmPP2C genes by long-distance signaling seem to be preliminary and require further clarification.
Comments:
1) Major concern of this study is that the design of a split-root experiment is not clear. Please, provide photos and/or scheme illustrating this experiment. In which part of the root system the expression levels have been analyzed? At which stage of the experiment the shoot was removed? How many days before gene expression analysis was the shoot cut off? How many biological repeats were used in this experiment for each treatment? How many times the experiment was repeated?
2) How many biological repeats were used for qPCR analysis in nitrate treatment experiments? How many times the experiments with nitrate treatment were repeated?
3) In the capture to Figure 1 it is mentioned that “Red, green, blue, and purple schemes represent maize, rice, Arabidopsis, and tomato, respectively”. However, this colors and corresponding gene names are not distinguished in Figure 1.
4) The authors should discuss the possible functions of the conserved domains found in ZmPP2C proteins. What are known about the domains found in PP2C proteins in other species?
5) It is not clear, why did the authors choose these particular six genes (ZmPP2C37, -41, -54, -69, -81, and -85) for the detailed analysis? The authors mentioned that “these showed potential responsiveness to nitrogen long-distance signals in roots”? How was it determined?
For example, author mentioned that ZmPP2C19, ZmPP2C37, and ZmPP2C71 contained GCN4 motif in their promoters, which suggest the involved of these genes in nitrogen stress response. However, ZmPP2C19 and ZmPP2C71 were not chosen for qPCR analysis. Moreover, ZmPP2C41,
ZmPP2C50, and ZmPP2C75 were identified as the closest paralogs of AtCEPH; however, among them, only ZmPP2C41 was analyzed by qPCR. What were the criteria for selecting these six genes for analysis?
6) Many nitrate-regulated genes were shown to have nitrate-response elements (NRE) in their promoters, known to be recognized by NLP transcription factors. Do ZmPP2C genes have such regulatory motif in their promoters?
7) In the capture to Figure 8, it is mentioned that “different letters indicate statistically significant differences (p < 0.05)”. However, еhere are no letters shown in the Figure 8.
8) Figure 6 has low resolution, and the text is not readable.
9) There are grammar errors (especially those related to the use of prepositions)
10) There are grammar errors (especially those related to the use of prepositions), as well as inaccuracies in wording and terms (for example, “..the results showed that the gene was located in the nucleus (line 345)”) throughout the text that should be fixed.
The quality of English should be improved.
Author Response
Response to Reviewer 1:
The present study focuses on the PP2C family in Zea mays. Authors identified 103 ZmPP2C genes in maize genome and estimated their expression levels based on available transcriptomic data. Moreover, for six ZmPP2C genes, the expression levels were analyzed by qPCR in the roots under different nitrogen status. Overall, the authors performed comprehensive analysis on identification and characterization of ZmPP2C genes. However, the results on regulation of A genes by long-distance signaling seem to be preliminary and require further clarification.
Point 1. Major concern of this study is that the design of a split-root experiment is not clear. Please, provide photos and/or scheme illustrating this experiment. In which part of the root system the expression levels have been analyzed? At which stage of the experiment the shoot was removed? How many days before gene expression analysis was the shoot cut off? How many biological repeats were used in this experiment for each treatment? How many times the experiment was repeated?
Response 1: Thank you for providing valuable comments on our manuscript. We have provided supplementary Figure S6 for a specific demonstration of the split-root experiment. In this experiment, we split the roots when the maize grew to 5 leaves and removed the aboveground part. After two days of treatment, the root on the left was sampled and analyzed quantitatively; in this experiment, three biological repetitions were set. Meanwhile, in the early stage of the experiment, we carried out multiple pre-experiments of root division and removal of shoots to ensure that the growth status of the plants was normal under the two treatments, which can be used for follow-up experiment. The specific experimental method has been revised in 5.9 (lines 477 to 497).
Point 2. How many biological repeats were used for qPCR analysis in nitrate treatment experiments? How many times the experiments with nitrate treatment were repeated?
Response 2: In the nitrogen-deficiency (0 mM) and high-nitrogen (15 mM) treatment experiments at different time points, we conducted three biological replicates with three plants in each replicate. The specific content was revised in Experimental Method 5.8 and the corresponding content (lines 474 to 476).
Point 3. In the capture to Figure 1 it is mentioned that “Red, green, blue, and purple schemes represent maize, rice, Arabidopsis, and tomato, respectively”. However, this colors and corresponding gene names are not distinguished in Figure 1.
Response 3: We have modified Figure 1.
Point 4. The authors should discuss the possible functions of the conserved domains found in ZmPP2C proteins. What are known about the domains found in PP2C proteins in other species?
Response 4: We have added an analysis of the conserved domain of ZmPP2Cs in the Discussion section (lines 329 to 332).
Point 5. It is not clear, why did the authors choose these particular six genes (ZmPP2C37, -41, -54, -69, -81, and -85) for the detailed analysis? The authors mentioned that “these showed potential responsiveness to nitrogen long-distance signals in roots”? How was it determined?
For example, author mentioned that ZmPP2C19, ZmPP2C37, and ZmPP2C71 contained GCN4 motif in their promoters, which suggest the involved of these genes in nitrogen stress response. However, ZmPP2C19 and ZmPP2C71 were not chosen for qPCR analysis. Moreover, ZmPP2C41, ZmPP2C50, and ZmPP2C75 were identified as the closest paralogs of AtCEPH; however, among them, only ZmPP2C41 was analyzed by qPCR. What were the criteria for selecting these six genes for analysis?
Response 5: We are very grateful for your suggestions. First of all, we selected six candidate genes, mainly for the following reasons: ZmPP2C54 and ZmPP2C69 may respond to long-distance nitrogen signaling (Ref 49), as demonstrated in the published data of China National Agricultural University Peng and discussed in the manuscript (lines 349 to line 355); secondly, based on the results of phylogenetic tree analysis, AtCEPH orthologs were found, as well as the nitrogen-related ortholog gene ZmPP2C85 reported in sorghum; meanwhile, we found ZmPP2C81 through the data in the public transcriptome database (Ref 50), combined with the selection of six genes that may be related to nitrogen stress from several parts of the data, the quantitative analysis of the six genes corresponding to different time treatments and responses to long-distance signals was carried out.
There is indeed a problem of incomplete consideration in the selection of genes. Therefore, based on your suggestion, we have treated ZmPP2C19, ZmPP2C71, ZmPP2C50, and ZmPP2C75 at different time points to improve the experimental design and quantitative analysis. At the same time, the other two genes, ZmPP2C50 and ZmPP2C75, which are the closest relatives to AtCEPH, were subjected to quantitative analysis of split-root and head-removal experiments in order to better explain the experimental results. The details are described in the research results.
Point 6. Many nitrate-regulated genes were shown to have nitrate-response elements (NRE) in their promoters, known to be recognized by NLP transcription factors. Do ZmPP2C genes have such regulatory motif in their promoters?
Response 6: In our analysis, the 103 ZmPP2C genes did not contain nitrate-response elements (NRE). Moreover, we did not find that the regulatory element was included in the reported species after reviewing the reference.
Point 7. In the caption of Figure 8, it is mentioned that “different letters indicate statistically significant differences (p < 0.05)”. However, there are no letters shown in the Figure 8.
Response 7: We have changed the legend, e.g., line 284.
Point 8. Figure 6 has low resolution, and the text is not readable
Response 8: We have modified Figure 6.
Point 9. There are grammar errors (especially those related to the use of prepositions)
Response 9:We reviewed the whole manuscript to rectify the grammatical errors and used one of the editing services listed at https://www.mdpi.com/authors/english to improve the quality of the manuscript.
Point 10. There are grammar errors (especially those related to the use of prepositions), as well as inaccuracies in wording and terms (for example, “the results showed that the gene was located in the nucleus (line 345)”) throughout the text that should be fixed.
Response 10:We have reviewed the whole manuscript to rectify the grammatical errors and used one of the editing services listed at https://www.mdpi.com/authors/english to improve the writing quality of the manuscript. Meanwhile, unclear expressions in the manuscript were corrected, e.g., lines 310 to 317 and lines 375 to 385.
Reviewer 2 Report
The manuscript is well written with the below minor corrections to be made:
(1) The Discussion part of the manuscript to be elaborated citing the relevance of the obtained results. As most of the results obtained are not discussed in the manuscript
(2) Present Conclusion at the end of the manuscript presenting the significant findings and its future prospects.
Author Response
Response to Reviewer 2:
The manuscript is well written with the below minor corrections to be made:
Point 1. The Discussion part of the manuscript to be elaborated citing the relevance of the obtained results. As most of the results obtained are not discussed in the manuscript.
Response 1: Thank you for providing valuable comments on our manuscript. We have revised this section, e.g., lines 310 to 317, lines 322 to 328, and lines 329 to 332.
Point 2. Present Conclusion at the end of the manuscript presenting the significant findings and its future prospects.
Response 2: We added a “Conclusions” section at the end of the manuscript (lines 386 to 395).
Reviewer 3 Report
The authors conducted comprehensive analysis of the maize protein phosphatase 2C (PP2C) family using various bioinformatic tools. However, the manuscript suffers from two major issues that are impeding its publication:
1) The manuscript cited prior works (ref. 36 and ref. 37) that have already reported similar findings. Specifically, Wei and Pan have already identified 104 PP2C members in maize. The authors need to clarify the original contributions of their research compared to these existing studies.
2) The authors propose a potential role of PP2C in long-distance signaling, but the conclusions lack substantial supporting data. The inclusion of cis-elements and expression analysis appears to be speculative and insufficient. To strengthen the manuscript's conclusions, the authors should conduct additional experiments offer more solid and data-driven insights.
There are a large number of grammar errors and formatting issues throughout the manuscript. The authors need to go through the manuscript carefully to correct those simple errors and improve the readability.
Author Response
Response to Reviewer 3:
The authors conducted comprehensive analysis of the maize protein phosphatase 2C (PP2C) family using various bioinformatic tools. However, the manuscript suffers from two major issues that are impeding its publication:
Point 1. The manuscript cited prior works (ref. 36 and ref. 37) that have already reported similar findings. Specifically, Wei and Pan have already identified 104 PP2C members in maize. The authors need to clarify the original contributions of their research compared to these existing studies.
Response 1: Thank you for providing valuable comments on our manuscript. Although Wei and Pan have already researched PP2C, this paper details further in-depth research based on two principles: First: Wei and Pan use the hidden Markov model (HMM) model to search for PP2C sequences. After screening the domain to confirm the PP2C candidate genes, this study used the reported Arabidopsis sequence to perform Blast and then used the structural domain to further determine the PP2C member after taking the intersection of the Blast and HMM results. The screening results may be more reliable. Based on Wei and Pan's research, we researched the integration of Arabidopsis, rice, and tomato in the evolutionary analysis, and at the same time, explored the three species of Arabidopsis, rice, and maize at the time of gene duplication, which is more capable of multi-dimensional maize evolutionary analysis, etc. In the Discussion section, we modified and added some content to improve readers’ understanding (lines 309 to 317).
Point 2. The authors propose a potential role of PP2C in long-distance signaling, but the conclusions lack substantial supporting data. The inclusion of cis-elements and expression analysis appears to be speculative and insufficient. To strengthen the manuscript's conclusions, the authors should conduct additional experiments offer more solid and data-driven insights.
Response 2: The discovery of the role of ZmPP2C in long-distance signals in maze proposed in this study is very novel, and we want to report it as soon as possible; at the same time, we used the split-root experiment and head-off experiment to verify the positive and negative, which is a relatively classic experiment and relatively reliable. The maize transgenic cycle is very long and cannot be completed quickly, and this study is a complete story by itself.
Round 2
Reviewer 1 Report
The authors responded to the reviewer's comments and revised the manuscript. I no longer have any comments, the manuscript can be recommended for publication.
Author Response
Thank you very much for your valuable comments.
Reviewer 3 Report
I think the authors did not adequately address the concern I raised, nor did they delve into the other relevant reference that has undertaken similar work. Fan et al. have already conducted a comprehensive analysis of the PP2C subfamilies utilizing remarkably similar methodologies. For instance, they employed the identical Pfam PP2C domain (PF00481) for their search. Moreover, Fan et al. searched PP2C homologs across 7 distinct species, including maize, rice, and Arabidopsis. They also did the Ka/Ks analysis.
It's perplexing to see a replication of the bioinformatic analysis utilizing conventional tools, unless novel insights have been garnered from such an endeavor. If the authors perceive the discovery of ZmPP2C's role in long-distance signaling as novel, it would be judicious to present this aspect as a distinct finding. However, the existing data does not present a compelling case. The pivotal experiment, as in Figure 8, demonstrated the upregulation of certain ZmPP2C genes under heterogeneous nitrogen treatment. Notably, considering the exceedingly low expression levels of these genes (as indicated by relative expression levels), augmenting the dataset from alternate angles to corroborate these changes would be advantageous—especially in the case of PP2C54 and PP2C85. Even in instances where two genes displayed upregulation, their relative expression levels hovered around 0.01?—prompting thinking on the physiological significance of such subtle fluctuations.
The discussion section should have an in-depth discussion of the potential involvement of these genes in long-distance signaling, if indeed the authors' assertion of novelty holds. Also, the linkage between the subcellular localization of PP2C85 and its role in long-distance signaling appears weak and requires further elucidation.
Minor editing of English language required
Author Response
Point 1: I think the authors did not adequately address the concern I raised, nor did they delve into the other relevant reference that has undertaken similar work. Fan et al. have already conducted a comprehensive analysis of the PP2C subfamilies utilizing remarkably similar methodologies. For instance, they employed the identical Pfam PP2C domain (PF00481) for their search. Moreover, Fan et al. searched PP2C homologs across 7 distinct species, including maize, rice, and Arabidopsis. They also did the Ka/Ks analysis.
It's perplexing to see a replication of the bioinformatic analysis utilizing conventional tools, unless novel insights have been garnered from such an endeavor. If the authors perceive the discovery of ZmPP2C's role in long-distance signaling as novel, it would be judicious to present this aspect as a distinct finding. However, the existing data does not present a compelling case. The pivotal experiment, as in Figure 8, demonstrated the upregulation of certain ZmPP2C genes under heterogeneous nitrogen treatment. Notably, considering the exceedingly low expression levels of these genes (as indicated by relative expression levels), augmenting the dataset from alternate angles to corroborate these changes would be advantageous—especially in the case of PP2C54 and PP2C85. Even in instances where two genes displayed upregulation, their relative expression levels hovered around 0.01? prompting thinking on the physiological significance of such subtle fluctuations.
Response 1: We sincerely appreciate your invaluable comments, which holds the potential to enhance the quality of our manuscript.
In previous studies, Wei and Pan meticulously characterized the protein phosphatase superfamily in maize, including 104 members in the ZmPP2C family, based on the maize B73 RefGen_v2 annotation; Fan et al. comprehensively analyzed the evolutionary dynamics and expansion patterns of 97 ZmPP2C family members, utilizing the maize B73 RefGen_v3 annotation. In our prior trial, we translated the gene models of the older version genomes to the newest B73-GRAMENE-NAM-5.0 genome. Interestingly, the 104 B73 RefGen_v2 genes in Wei and Pan’s study and the 97 B73 RefGen_v3 genes in Fan et al.’s study matched to 121 and 110 gene models in the B73-GRAMENE-NAM-5.0 genome. Only 91 and 85 were shared with our study. Notably, among these genes that were not shared, at least 14 genes did not emerge in our BLAST analysis, conducted for sequence similarity determination. It remains uncertain whether these disparities stem from substantial variations across genome assembly and annotation versions, or if they can be attributed to the absence of sequence similarity analysis via BLAST in the two preceding studies. We did not report these distinctions in our manuscript for the reason that our research goal was not to compare these studies.
To validate the potential involvement of these PP2C candidate genes in long-distance signaling, we employed two classic physiological methodologies widely recognized for the study of long-distance signaling, namely the split-root experiment and shoot removal experiment. Notably, numerous scholars have successfully corroborated the existence of long-distance signals through the utilization of these methods (Ref 50 and Ref 51). Our previous studies (Ref 52 and Ref 53) have established that the shoot plays a central role in long-distance signaling. In this study, the split-root experiment indicated downregulation of ZmPP2C54 and ZmPP2C85 under heterogeneous low nitrate conditions. To confirm, we conducted a shoot removal experiment, which eliminated the negative regulation of these genes. These experiments provide strong evidence for the involvement of these genes in long-distance signaling.
Regarding the value "0.01," it's a relative fold change compared to the highly stable reference gene ZmElF1. We've updated this value by normalizing relative gene expression to 1 in the CK-HomoHN treatment for greater accuracy.
Point 2: The discussion section should have an in-depth discussion of the potential involvement of these genes in long-distance signaling, if indeed the authors' assertion of novelty holds. Also, the linkage between the subcellular localization of PP2C85 and its role in long-distance signaling appears weak and requires further elucidation.
Response 2: Thank you for the concerns. We have incorporated a discussion of these points within the manuscript (Lines 359 to 369). The subcellular localization of ZmPP2C85 in the nucleus is different from the localization of AtCEPH, which was localized in both nucleus and cytoplasm of root epidermal cells. However, these differences might at least partly be attributed to the differences of experimental system. Hence, we cannot dismiss the potential contribution of this gene to long-distance signaling. However, we fully agree with the reviewer's point that the linkage between the subcellular localization of PP2C85 and its role in long-distance signaling need to be further illustrate in the following up experiments (Lines 381 to 386).
Round 3
Reviewer 3 Report
The authors have addressed my concerns in some sort.
Minor editing of English language required.